chemical engineering/energy

falling film, model, numerical simulation, heat transfer coefficient

**Author for correspondence:**
Jing Fang
e-mail: ctstfj@163.com

# Establishment of the falling film evaporation model and correlation of the overall heat transfer coefficient

## Jing Fang, Kaixuan Li and Mengyu Diao

Chemical Engineering Department, Hebei University of Technology, Tianjin 300132, People's Republic of China

 JF, 0000-0002-9327-8735; KL, 0000-0001-6057-5066

In order to study the heat transfer of the falling film evaporator with phase change on both sides, in this paper we built the mathematical model and the physical model where the liquid film inside the tube is laminar and turbulent. The film thickness of the condensate at different axial positions, total condensate volume and velocity distribution, and temperature distribution of condensate outside the tube can be obtained by calculating the proposed model. Meanwhile, the liquid film thickness, velocity distribution and temperature distribution inside the tube were obtained by numerical simulation by considering the influence of the liquid film with different compositions on the heat transfer during fluid flow. With ethanol–water as the system, the overall heat transfer coefficient and heat transfer quantity of the falling film evaporator were obtained by the calculation of the model. The accuracy of the proposed model was confirmed by experiments. The model and the calculation of heat transfer proposed in this paper have enormous significance for the basic data and theoretical guidance of the heat transfer performance prediction and operational optimization of the evaporator.

## 1. Introduction

The falling film evaporation has widely concerned various scholars for its short residence time, small temperature difference and high heat transfer coefficient. It has likewise become a hot topic in the field of heat transfer research. Further exploring the phenomenon of falling film flow has great significance for the research of enhanced heat transfer. Thermal properties are critical for enhanced heat transfer [1–5]. The research on the heat transfer of falling film evaporation is categorized into three: theoretical research, experimental research and numerical simulation (table 1).

**Table 1.** Nomenclature.

| nomenclature | | greek symbols | |
| --- | --- | --- | --- |
| $c_p$ | heat capacity of condensate, J kg$^{-1}$ K$^{-1}$ | $\delta$ | liquid film thickness, m |
| $G$ | gravity acceleration, m s$^{-2}$ | $\mu$ | condensate viscosity, Pa s |
| $H$ | heat transfer coefficient, W m$^{-2}$ K$^{-1}$ | $\sigma$ | surface tension, N m$^{-1}$ |
| $h_0$ | condensation phase change heat transfer coefficient, W m$^{-2}$ K$^{-1}$ | $\Gamma$ | wetting rate, kg m$^{-1}$ s$^{-1}$ |
| $h_i$ | evaporation phase change heat transfer coefficient, W m$^{-2}$ K$^{-1}$ | $\tau$ | the minimum wetting rate of the evaporation tube, kg m$^{-1}$ s$^{-1}$ |
| $K$ | overall heat transfer coefficient, W m$^{-2}$ K$^{-1}$ | | |
| $k_w$ | thermal conductivity of the tube wall, W m$^{-2}$ K$^{-1}$ | | |
| $k_o$ | thermal conductivity of the condensate, W m$^{-2}$ K$^{-1}$ | | |
| $k_i$ | thermal conductivity of the raw material, W m$^{-2}$ K$^{-1}$ | | |
| $m_0$ | imported liquid mass flow rate, kg s$^{-1}$ | | |
| $Pr$ | Prandtl number of condensate | | |
| $q_w$ | interface heat flow rate, W m$^{-1}$ | | |
| $Re_l$ | in-tube liquid film Reynolds number | | |
| $Re_c$ | outer tube condensate Reynolds number | | |
| $R_i$ | the inner radius of the tube, m | | |
| $R_0$ | the outer radius of the tube, m | | |
| $R$ | latent heat of condensation, kJ kg$^{-1}$ | | |
| $T_s$ | steam-saturation temperature, °C | | |
| $T_w$ | inner wall temperature of the inner tube, °C | | |
| $T_{ow}$ | outer wall temperature of the outer tube, °C | | |
| $T_{si}$ | ethanol–water solution liquid film outer layer temperature, °C | | |
| $T_{so}$ | condensate outer layer temperature, °C | | |
| $u_x$ | axial velocity, m s$^{-1}$ | | |
| $u_m$ | body flow rate, m s$^{-1}$ | | |
| $V$ | radial speed, m s$^{-1}$ | | |
| $W$ | the condensate per unit time on the outer wall of the tube, kg s$^{-1}$ | | |

In 1916, Nusselt [6] revealed the liquid film flow regular pattern and laid the foundation for the heat transfer research for the smooth vertical tube surface. However, there is a great difference between this ideal flow process and the actual application. There is a substantial error between the predicted result and the experimental value. The result and the experimental value are larger than the theoretical value.

Colburn & Hougen [7] proposed the basic theory of heat and mass transfer, pointing out that the energy conservation equation and the mass transfer equation have similar forms. The heat transfer correlation can be used to calculate mass transfer and confirm that the performances of heat transfer and mass transfer are almost in the same order of magnitude. Whitman [8] proposed the two-film theory, but it did not consider the mass transfer effect, and the phase interface is not strictly thermodynamic equilibrium.

Colburn [9] assumes that under the condition of no tangential shear stress at the vapour–liquid interface, the critical Reynolds number $Re_{cr}$ of the liquid film flow from the laminar to turbulent flow is close to 2000 in the full tube flow, and a semi-empirical relationship is proposed. At present, there is no uniform standard for the mechanism of the influence of momentum, heat and mass transfer on the liquid film. Lorentz & Yung [10] and Gambaryan-Roisman & Stephan [11] dealt with the study of heat transfer enhancement in falling film on the ribbed surface, but there are almost no systematic data. Struve [12] studied the liquid film in the steam heating tube and obtained the correlation under the corresponding conditions. Herbert & Sterns [13] experimentally studied the falling film evaporation on the resistance wire heating inner tube liquid film and obtained the analogous correlation. Chun & Seban [14] and Fujita & Ueda [15] conducted experimental studies on the liquid film heated by the electric heating tube and corresponding correlation, respectively. Schnabel & Schlunder [16] analysed the experimental data of several researchers to obtain the heat transfer coefficient correlation. Shah [17–20] analysed the 136 datasets from 67 sources to obtain the scope of each heat transfer correlation.

Evaporation occurs at the surface of the liquid film when the multi-component liquid film receives heat during the falling process. But when the temperature of the tube wall contacted by the liquid film is higher than its own boiling point, a relatively sharp nuclear boiling occurs on the liquid–solid surface. Dhir [21,22] performed a numerical simulation study on the nucleate boiling state of the liquid film. However, since the flow of the liquid film is extremely unstable under boiling conditions and it is difficult to discuss, research in this field is not sufficient. Hoke & Chen [23] obtained the mass transfer coefficient of the evaporation process of the binary mixture by the momentum equation and the composition equation simultaneously. However, the energy equation is not seen in the model, the influence of temperature is neglected and the obtained mass transfer coefficient is not accurate.

The studies only on a single component ignore the mass transfer of multi-component fluids. Most studies focus only on unilateral heat transfer of the vertical tube, while the falling film evaporation is bilateral heat transfer. The error between the obtained heat transfer coefficient and the actual is large. Therefore, research on the bilateral heat transfer of falling film evaporation needs to be solved, which will provide theoretical guidance for the design of falling film evaporator.

In this paper, the mathematical model and the physical model were established. The thickness distribution, velocity distribution and temperature distribution of the liquid film inside and outside the tube were obtained by numerical simulation. The calculation formula of the overall heat transfer coefficient of the falling film evaporator was obtained and verified by experiments.

# 2. Heat transfer model

The saturated water vapour is used to heat the saturated ethanol–water solution which is in the tube. Different components in ethanol–water will evaporate under different degrees at different tube lengths. Therefore, the actual evaporation of the solution is complicated. To simplify the model analysis, some minor factors are ignored, some assumptions are made, and the physical and mathematical models are set up.

As shown in figure 1, the fluid in the tube is the ethanol–water solution. The fluid outside the tube is water vapour, and the outer wall surface of the inner tube is water vapour condensate film. This section is meant for the phase change of the vertical tube. Based on the boundary layer theory, the physical and mathematical models of the liquid film are established by the Cartesian coordinate system. The liquid film velocity distribution, temperature distribution, film thickness and heat transfer characteristics on both sides of the vertical pipe were studied.

## 2.1. Out-of-tube falling film model

The fluid outside the falling film tube is saturated water vapour, and there is no mass transfer. The vapour changed to condensation liquid by releasing heat, so the equation is mainly established for the condensate, and the following assumptions are made for the condensate flow: (1) constant property; (2) ignore the subcooling and inertial force of the liquid film; (3) ignore the vapour density; (4) the surface of the liquid film is flat without fluctuation; (5) the gas flow rate is small; ignore its effect on the liquid film; and (6) the temperature inside the film is linearly distributed, and it is considered that heat transfer in the liquid film is only heated conduction.

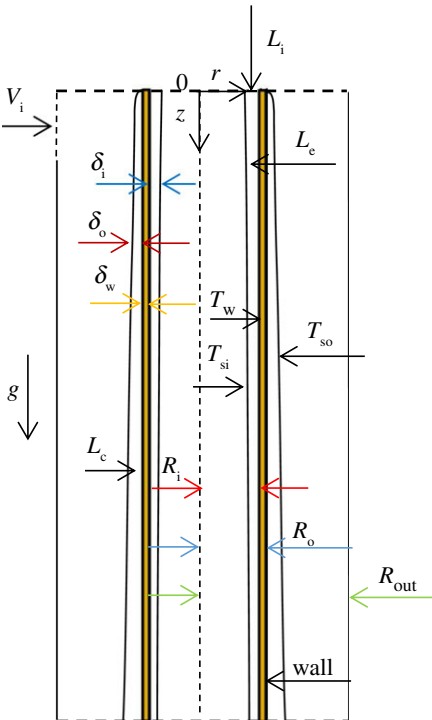

**Figure 1.** Diagram of falling film evaporation.

As shown in figure 2, the shell side saturated water vapour changed to condensation liquid by releasing heat, and the condensate flows downward along the outer surface of the vertical tube with a steady laminar liquid film. It is considered that the fluid flow is in the $x$-direction, the velocity distribution of the fluid is in the $y$-direction and the position of the liquid film immediately above the outer wall of the inner tube is $y = 0$, the density and viscosity of the fluid are $\rho$ and $\mu$, respectively.

### 2.1.1. Outer tube condensate velocity distribution

Continuity equation

$$\frac{\partial u}{\partial x} + \frac{\partial v}{\partial y} = 0. \tag{2.1}$$

The equation of motion

$$\rho_l \left( u \frac{\partial u}{\partial x} + v \frac{\partial u}{\partial y} \right) = -\frac{\mathrm{d}P}{\mathrm{d}x} + \rho_l g + \mu \frac{\partial^2 u}{\partial y^2}, \tag{2.2}$$

where the subscript l indicates the liquid phase.

According to hypotheses 3 and 4 (see §2.1) and equation (2.2), it can be reduced to

$$\rho_l g + \mu \frac{\partial^2 u}{\partial y^2} = 0. \tag{2.3}$$

Boundary conditions:

(1) $y = 0$: $u = 0$.

(2) $y = \delta$: $\left. \dfrac{\mathrm{d}u}{\mathrm{d}} \right|_{\delta} = 0$ (ignore the effect of gas on the liquid film; ignore the shear stress).

The axial velocity distribution of the steady laminar flow falling condensate can be obtained

$$u_x = \frac{\rho g}{2\mu} (2\delta y - y^2). \tag{2.4}$$

The main body flow rate

$$u_m = \frac{\rho g \delta^2}{3\mu}. \tag{2.5}$$

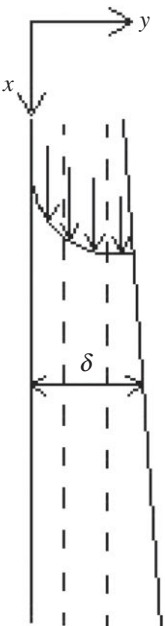

**Figure 2.** Flow of vapour–liquid multiphase outside the tube.

Then, the amount of condensate per unit time of the outer wall

$$w = \frac{\rho^2 g \delta^3}{3\mu}. \tag{2.6}$$

The condensate liquid film thickness

$$\delta = \left[ \frac{4\mu k_o (T_s - T_w) x}{\rho^2 g r} \right]^{1/4}. \tag{2.7}$$

The amount of condensate

$$w = \frac{8}{5} \pi R_o x^{5/4} \left[ \frac{4\mu k_o \rho^2 (T_s - T_w)}{g r} \right]^{1/4}, \tag{2.8}$$

where $k_o$ is the thermal conductivity of the condensate, W m$^{-1}$ K$^{-1}$. According to the numerical solution of equations (2.4) and (2.8), figure 3 is obtained. When $x = 0$ and $y = 0$: $u = 0$, $T_s = 373.15$ K, $T_w = 371.15$ K. It can be seen from the figure that in the $x$-direction, the liquid film speed is getting larger and larger. The liquid film on the outer wall of the inner tube gradually increases from wall to outer direction; however, the gradient of the liquid film thickness gradually decreases as $x$ increases.

### 2.1.2. Outer tube condensate temperature distribution

Energy equation

$$u_x \frac{\partial T}{\partial x} + u_y \frac{\partial T}{\partial y} = \alpha \frac{\partial^2 T}{\partial y^2}. \tag{2.9}$$

The above equation can be simplified according to hypothesis 6 (see §2.1) as

$$\alpha_l \frac{\partial^2 T}{\partial y^2} = 0. \tag{2.10}$$

Boundary conditions:

(1) $y = 0$: $T = T_{ow}$ (the film temperature at the wall is considered as the wall temperature).
(2) $y = \delta$: $T = T_{so}$ (liquid film outer layer temperature is regarded as the steam temperature).

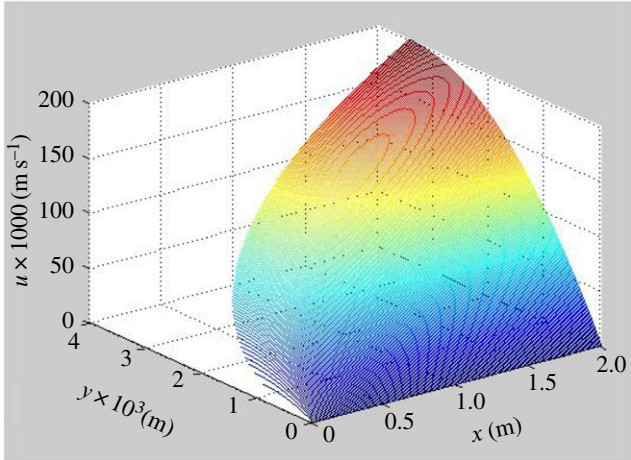

**Figure 3.** Axial velocity distribution of the condensate outside the tube.

The axial temperature distribution of the falling condensate can be obtained when the steady laminar flow is obtained

$$T = T_{iw} + \frac{y}{\delta}(T_{so} - T_{ow}).$$ (2.11)

Figure 4 shows the temperature distribution of the condensate outside the tube. When $x = 0$ and $y = 0$, $T = 371.15$ K; when $y = \delta$, $T = 373.15$ K. It can be seen from the figure that the liquid film temperature of the outer wall of the inner tube gradually increases from the wall surface to the outer layer, and the outermost layer temperature of the liquid film is the highest.

## 2.2. In-tube falling film model

In order to make the model easy to solve and conform to the engineering practice, some minor factors are neglected, and the following assumptions are made:

(1) The liquid film flow is considered to be laminar and incompressible, and the maximum $Re$ is less than 1500. The Reynolds number of the liquid film is calculated by the standard definition $Re_l = 4m_0/\mu \cdot 2\pi r$, where $m_0$ is the mass flow rate of liquid phase inlet, kg s$^{-1}$.
(2) When the flow is turbulent, only the laminar inner layer and the turbulent layer are considered. When the liquid film flow is in the turbulent flow, only the velocity, temperature and concentration distribution in the inner layer of the laminar flow are considered. Among them, the relationship between the thickness of the laminar sublayer and the liquid Reynolds number is $\delta/d = 61.5/Re^{7/8}$, where $d$ is the inner diameter of the tube, in m.
(3) It is assumed that evaporation of the liquid film occurs only at the interface, and phase equilibrium is reached at the vapour–liquid interface.
(4) Loss of liquid film heat caused by axial temperature is negligible.
(5) Consider the liquid film properties as a constant. The saturation temperature of the liquid film at a certain pressure is the qualitative temperature.
(6) Assuming that the flash steam is stationary, there is no pressure drop in the tube, and the interface temperature of the liquid film is approximately considered to be the flash steam temperature.

Figure 5 shows the vapour–liquid multiphase flow in the tube. The vapour–liquid multiphase flow is a steady flow in the tube and along the $x$-direction, the velocity distribution changes along the $y$-direction.

### 2.2.1. Distribution of liquid film velocity in the tube

In the steady state, the continuity equation of the liquid film in the tube

$$\frac{\partial u}{\partial x} + \frac{\partial v}{\partial y} = 0.$$ (2.12)

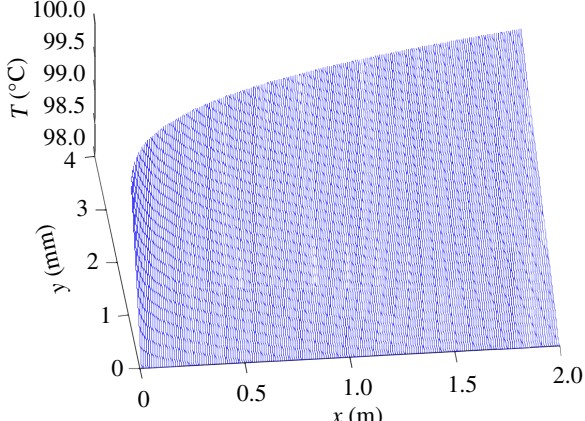

**Figure 4.** Tube condensate temperature profile.

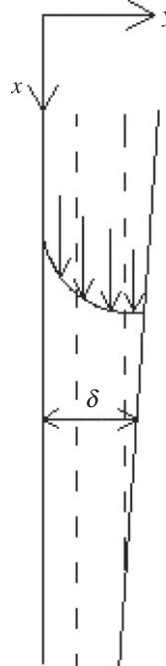

**Figure 5.** Flow of vapour–liquid multiphase in the pipe.

The equation of motion

$$\rho_l\left(u\frac{\partial u}{\partial x} + v\frac{\partial u}{\partial y}\right) = -\frac{\mathrm{d}P}{\mathrm{d}x} + \rho_l g + \mu\frac{\partial^2 u}{\partial y^2}. \tag{2.13}$$

Simplified

$$\rho_l g + \mu\frac{\partial^2 u}{\partial y^2} = 0. \tag{2.14}$$

Boundary conditions:
(1) $y = 0$: $u = v = 0$.
(2) $y = \delta$: $\left.\dfrac{\mathrm{d}u}{\mathrm{d}y}\right|_\delta = 0$ (laminar flow)   and   $y = \delta_b$: $\left.\dfrac{\mathrm{d}u}{\mathrm{d}y}\right|_{\delta_b} = 0$ (laminar sublayer).

The axial velocity distribution of the fluid in steady laminar flow can be obtained

$$u_x = \frac{\rho g}{2\mu}(2\delta y - y^2). \tag{2.15}$$

The Reynolds number can be abbreviated as

$$Re = \frac{4\Gamma}{\mu},$$ (2.16)

where $\Gamma$ is the wetting rate, kg m$^{-1}$ s$^{-1}$.

According to equations (2.12), (2.15) and the boundary conditions, the radial velocity can be obtained

$$v = -\frac{\rho g}{2\mu} y^2 \frac{\mathrm{d}\delta}{\mathrm{d}x}.$$ (2.17)

### 2.2.2. Liquid film temperature distribution in the tube

Energy equation

$$\frac{\partial T}{\partial \theta} + u_x \frac{\partial T}{\partial x} + u_y \frac{\partial T}{\partial y} + u_z \frac{\partial T}{\partial z} = \alpha \left( \frac{\partial^2 T}{\partial x^2} + \frac{\partial^2 T}{\partial y^2} + \frac{\partial^2 T}{\partial z^2} \right)$$ (2.18)

and

$$\alpha = \frac{k_i}{\rho c_p}.$$ (2.19)

In this study, the temperature change caused by the radial direction is ignored, and the energy equation can be reduced to

$$u_x \frac{\partial T}{\partial x} = \frac{k_i}{\rho c_p} \frac{\partial^2 T}{\partial y^2}.$$ (2.20)

Boundary conditions:

(1) $x = 0$: $T = T_i$.

(2) $y = 0$: $\frac{\partial T}{\partial} = -\frac{q_w}{k_i}$ (only heat conduction between liquid films).

(3) $y = \delta$: $T = T_{seq} = T_s''$ (the outer layer temperature of the liquid film is equal to the flash steam temperature).

The temperature distribution in the liquid film in the tube is obtained by assumptions 3 and 6 (see §2.2) and equations (2.18), (2.19) and (2.20) as shown in figure 6. In order to make the temperature change more clearly, the $y$-axis coordinates range from 0.3 to 0. When $x = 0$ and $y = 0$, $T = 346.15$ K; when $y = \delta$, $T = 342.15$ K.

It can be seen from figure 6 that the temperature of the liquid film is related to the film thickness. The temperature of the innermost layer of the liquid film is the highest, and the temperature gradient is also the largest at the innermost layer of the liquid film. The temperature fluctuation in the outer layer of the liquid film is small, and the thickness of the liquid film is not changed much.

# 3. Heat transfer coefficient correlation

The Nusselt number is a parameter indicating the intensity of convective heat transfer, which is defined as

$$Nu = \frac{hd}{k}.$$ (3.1)

For the liquid film in the laminar flow state, heat is conducted to the pipe wall through the liquid film, and the condensation (or evaporation) heat transfer quantity is the heat conduction amount of the liquid film layer of the microelement.

According to the Fourier formula, heat transfer tube inner wall surface temperature $T_{iw}$ can be obtained

$$T_{iw} = T_{ow} - \frac{q/k_w \cdot \ln R_0/R_i}{2\pi L}.$$ (3.2)

The heat transfer coefficient is expressed as

$$h = \frac{Q}{A\Delta T}.$$ (3.3)

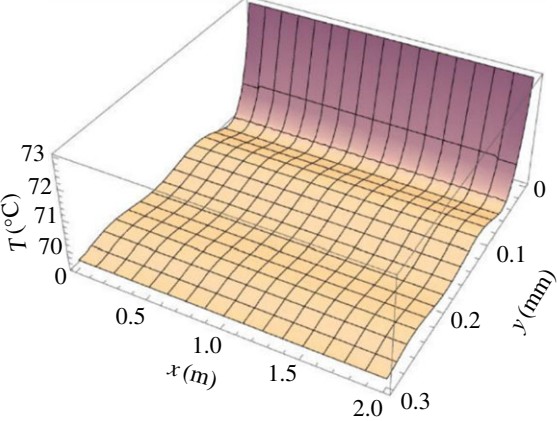

**Figure 6.** Tube temperature distribution.

Local heat transfer coefficient is obtained when the steam condenses (or the film evaporates) outside the pipe wall according to the Newton cooling law and the heat conduction equation

$$h = \frac{q_w}{T_1 - T_2} = \frac{k_o}{R \ln R / (R + \delta)}. \tag{3.4}$$

The numerical solution of the local heat transfer coefficient is obtained by the liquid film velocity distribution, temperature distribution and film thickness. The analytical solution of the heat transfer coefficient is fitted according to the numerical solution.

The heat transfer amount is obtained by calculating the phase variable, and the heat transfer coefficient is obtained.

## 3.1. Correlation of condensation heat transfer coefficient outside the tube

In this paper, the Reynolds number of the condensate outside the tube is in the laminar flow range. The Nusselt theory holds that the heat transfer coefficient of the condensed phase transition of saturated water vapour is mainly related to the physical properties and flow rate of the condensate. Therefore, the factors which affect the condensation heat transfer coefficient of the condensate laminar flow can be expressed as

$$h_o = f\left(\Gamma, \rho, \mu, k, c_p\right). \tag{3.5}$$

The Reynolds number and the Prandtl number of the condensate can be expressed as

$$Re_c = \frac{4\Gamma}{\mu} \tag{3.6}$$

and

$$Pr = \frac{c_p \mu_1}{k_o} = \frac{v}{\alpha}. \tag{3.7}$$

Therefore, by applying the dimensional analysis method, the following correlation can be obtained

$$h_o = aRe_c^m Pr^n. \tag{3.8}$$

Considering the influence of the parameters such as the Reynolds number and the Prandtl number on the condensation heat transfer coefficient, combining the heat transfer coefficient obtained in the model, the correlation coefficient between the heat transfer coefficient and the Nusselt number is obtained

$$h_o = 5.32 \times 10^4 \, Re_c^{-0.1418} Pr_1^{-3.1975} \tag{3.9}$$

and

$$Nu = 5.32 \times 10^4 \, Re_c^{-0.1418} Pr_1^{-3.1975} \frac{d_0}{k_w}. \tag{3.10}$$

## 3.2. Correlation of evaporation heat transfer coefficient in the tube

Saturated ethanol–water undergoes phase change by absorbing heat. It is considered that the flash steam has a downward force on the liquid film flow, which helps to enhance heat transfer. Considering various factors, the

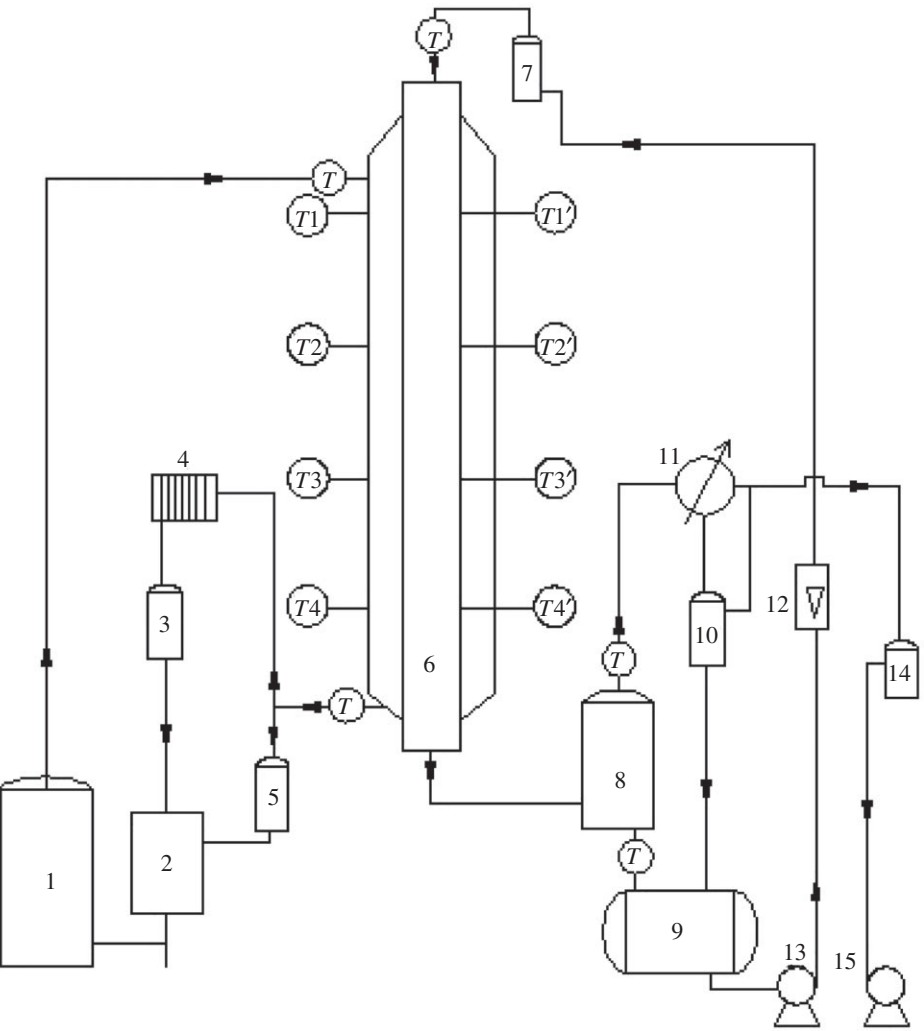

**Figure 7.** Falling film evaporation. 1, steam generator; 2, water tank; 4, air cooler; 6, falling film evaporator; 7, preheater; 8, vapour–liquid separation chamber; 9, raw material tank; 11, condenser; 12, glass rotameter; 13, raw material pump; 3, 5, 10, holding tank; 14, buffer tank; 15, vacuum pump.

evaporation heat transfer coefficient is related to the Reynolds number of the saturated liquid, the Reynolds number of the flash steam and the physical properties of the vapour and liquid. Therefore, the factors affecting the evaporation heat transfer coefficient of the saturated ethanol–water liquid can be expressed as

$$h_i = f(\Gamma, \rho, \mu, k, c_p). \tag{3.11}$$

The Reynolds number and the Prandtl number of the saturated liquid can also be expressed by equations (3.6) and (3.7)

$$Re_v = \frac{4V}{\pi(d - 2\delta)\mu_v}. \tag{3.12}$$

Therefore, by applying the dimensional analysis method, the following correlation can be obtained

$$h_i = aRe_l^m Re_v^n. \tag{3.13}$$

The heat transfer coefficient, the saturated liquid and the Reynolds number of the flash steam are fitted to the quasi-correlation coefficient of the heat transfer coefficient. The correlation of the heat transfer coefficient is divided into the laminar flow and turbulent flow

$$h_i = 62.09Re_l^{-0.01239} Re_v^{0.3427} \quad \text{(laminar flow)} \tag{3.14}$$

and

$$h_i = 73.38Re_l^{0.0063} Re_v^{0.3113} \quad \text{(turbulent flow)}. \tag{3.15}$$

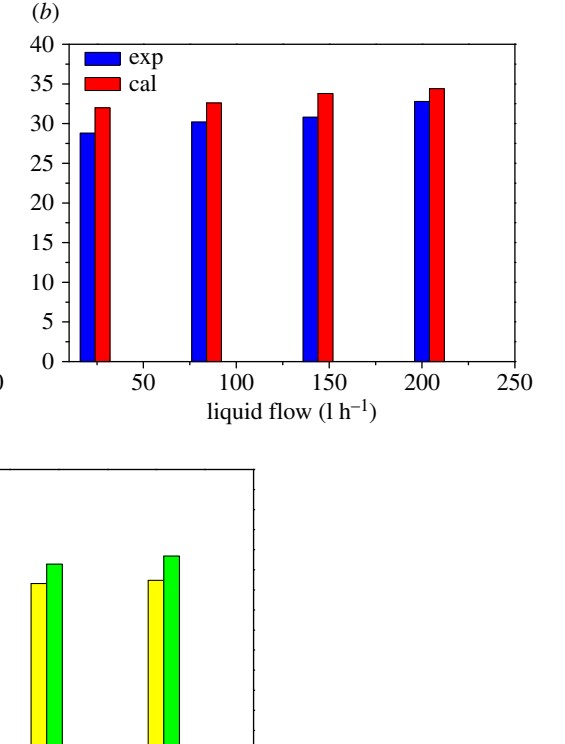

**Figure 8.** Comparison of the heat transfer with the pressure in the tube and the feed flow as variables. (*a*) $P = 50$ kPa; (*b*) $P = 60$ kPa; (*c*) $P = 70$ kPa.

**Table 2.** Equipment parameters and manipulated variables.

| variable | numerical value | | | |
|---|---|---|---|---|
| feeding flux (l h$^{-1}$) | 40 | 100 | 160 | 220 |
| absolute pressure inside the tube (kPa) | 50 | 60 | 70 | 80 |
| ethanol content (mass fraction) | 10% | 20% | 30% | — |
| evaporation tube size (mm) | outer diameter 81 | inner diameter 20 | tube length 2000 | |

The corresponding Nusselt number correlation

$$Nu = 62.09 Re_l^{-0.01239} Re_v^{0.3427} \frac{d_i}{k_w} \quad \text{(laminar flow)} \tag{3.16}$$

and

$$Nu = 73.38 Re_l^{0.0063} Re_v^{0.3113} \frac{d_i}{k_w} \quad \text{(turbulent flow).} \tag{3.17}$$

## 3.3. Correlation of overall heat transfer coefficient

$$\frac{1}{h} = \frac{1}{h_0} + \frac{b d_i}{k_w d_0} + \frac{d_o}{h_i d_i}. \tag{3.18}$$

According to the phase change heat transfer coefficient, it can be fitted into equation (3.18) to obtain the overall heat transfer coefficient of the falling film evaporator (figure 7). The results are shown in equations (3.19) and (3.20).

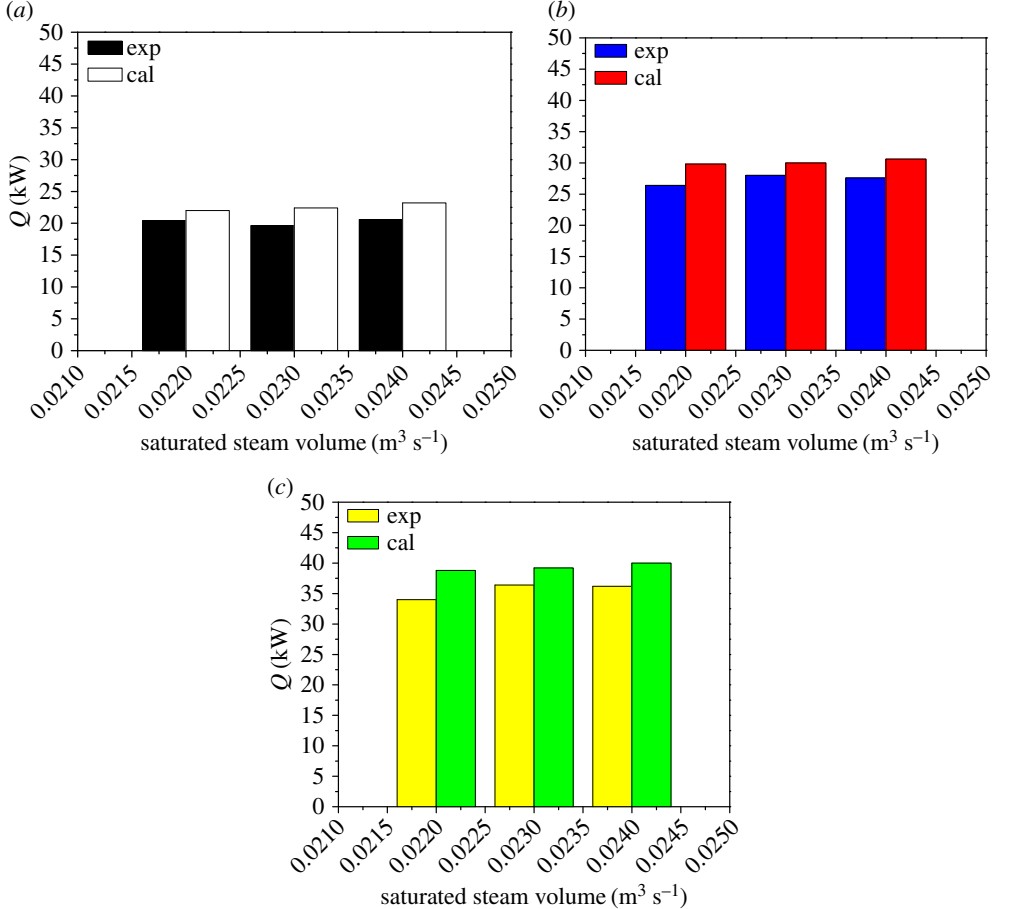

**Figure 9.** Comparison of heat transfer quantity with saturated vapour volume and ethanol content as variables. (*a*) 10% ethanol–water solution; (*b*) 20% ethanol–water solution; (*c*) 30% ethanol–water solution.

When the liquid film in the tube is laminar

$$\frac{1}{h} = 1.88 \times 10^3 Re_c^{0.1418} Pr^{3.1975} + \frac{bd_i}{k_w d_0} + 0.016 \frac{d_i}{d_0} Re_l^{0.01239} Re_v^{-0.3427}. \tag{3.19}$$

When the liquid film in the tube is turbulent

$$\frac{1}{h} = 1.88 \times 10^3 Re_c^{0.1418} Pr^{3.1975} + \frac{bd_i}{k_w d_0} + 0.0136 \frac{d_i}{d_0} Re_l^{-0.0063} Re_v^{-0.3113}. \tag{3.20}$$

where $Re_c$ is the Reynolds number of condensate outside the tube; $Pr$ is the Prandtl number of condensate outside the tube; $Re_l$ is the Reynolds number of the mixed liquid membrane in the tube; $Re_v$ is the Reynolds number of flash steam in the tube.

## 4. Model verification

In this experiment, in order to be close to industrial applications, the saturated ethanol–water in the tube was heated by the saturated water vapour. The heat transfer performance of the vertical tube evaporator was analysed. The main parameters for measurement are wall temperature of four corresponding positions inside and outside the evaporator heat exchange tube, liquid feed temperature, vapour and liquid temperature at the outlet of the inner tube, amount and composition of flash steam, feed temperature of saturated steam, the amount and temperature of the condensate, and the pressure inside the tube. The equipment parameters and manipulated variables are shown in table 2.

The effects of feed flow rate, temperature difference between the internal and external pipe, ethanol content and saturated steam heating on the heat transfer performance are investigated by theoretical calculation and experimental methods. The overall heat transfer obtained in the two models was

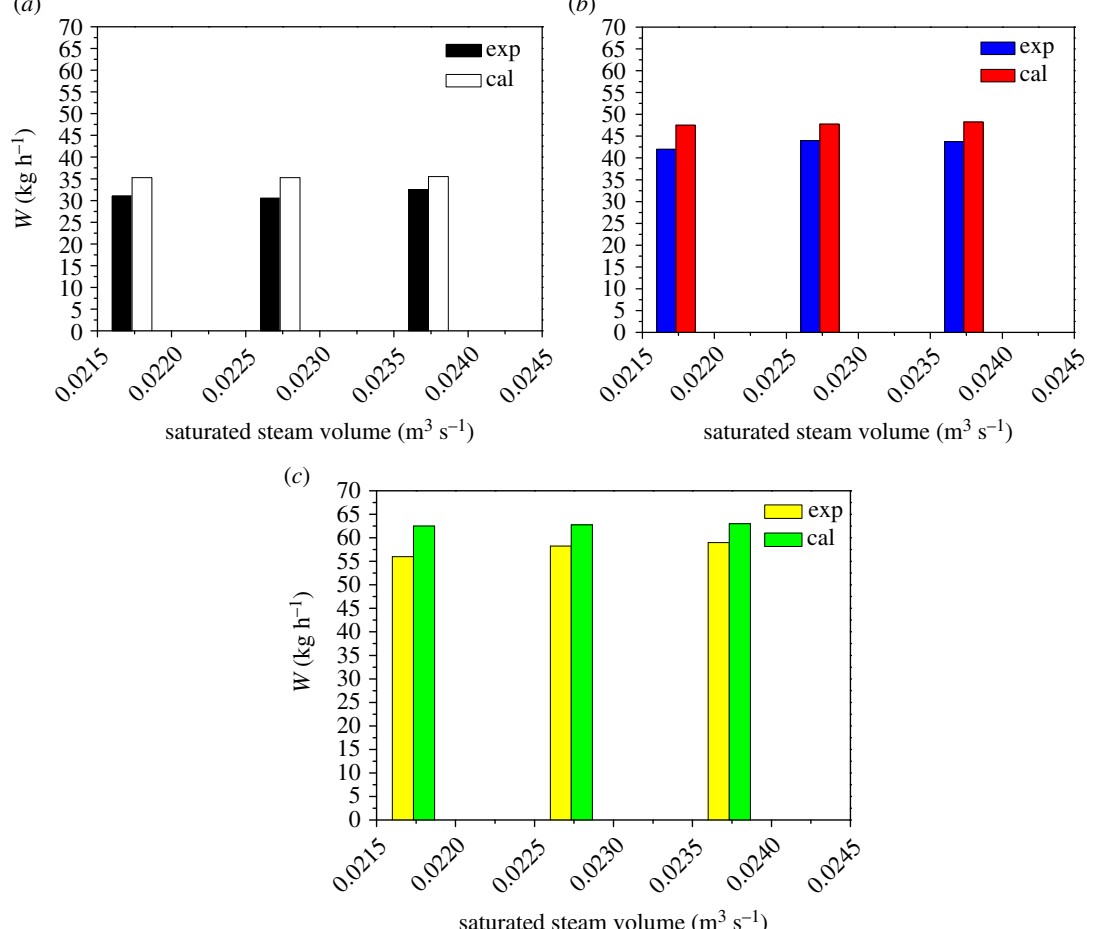

**Figure 10.** Comparison of condensate volume with saturated water vapour content and ethanol content as variables. (*a*) 10% ethanol – water solution; (*b*) 20% ethanol – water solution; (*c*) 30% ethanol – water solution.

compared. It can be found from figures 8 and 9 that the calculated values of the theoretical calculation are slightly larger than the experimental values. This phenomenon is due to the increasing thickness of the condensate outside the tube during falling, the thermal resistance of thickness hinders the heat transfer. The effects of the thermal resistance of thickness, the film thickness at different axial positions of the evaporating tube, the liquid film composition, heat transfer mode and liquid film temperature on the physico-chemical properties are all related.

It can be seen from figures 8 and 10 that as the feed flow rate increases, the amount of condensing of the steam outside the tube and the amount of heat transfer increase correspondingly. The main reason is that as the feed flow rate increases, the fluid turbulence increases and the heat transfer performance increases. As the pressure inside the heat transfer tube increases, the amount of steam condensation and heat transfer outside the tube decreases, mainly because the saturation temperature of the raw material increases with the pressure inside the heat transfer tube. The temperature difference between the inside and the outside becomes larger, the heat transfer driving force is increased, and the heat transfer effect is enhanced. It can be seen from figures 9 and 10 that as the concentration of ethanol in the ethanol mixture in the tube is increased, the amount of condensate collected in the shell is increased at a certain period of time. However, the change of saturated steam has less influence on the amount of condensate. This phenomenon can be explained by the fact that the ethanol content in the evaporation tube is increased, so that the average concentration of the solution is increased, which will increase the thermal conductivity of the liquid film, so that the heat transfer effect is enhanced, and thus, the amount of shell condensate exhibited will be increased. In addition, the effect of the increase in the liquid film flow is enough to offset the increase in the average temperature of the liquid film in the tube due to the increase in ethanol content and the heat transfer temperature. The disadvantages caused by the reduction are that under certain operating conditions, the heat required for evaporation of the liquid film in the evaporation tube is substantially constant, and the amount of saturated steam heating is increased without changing the heat transfer effect.

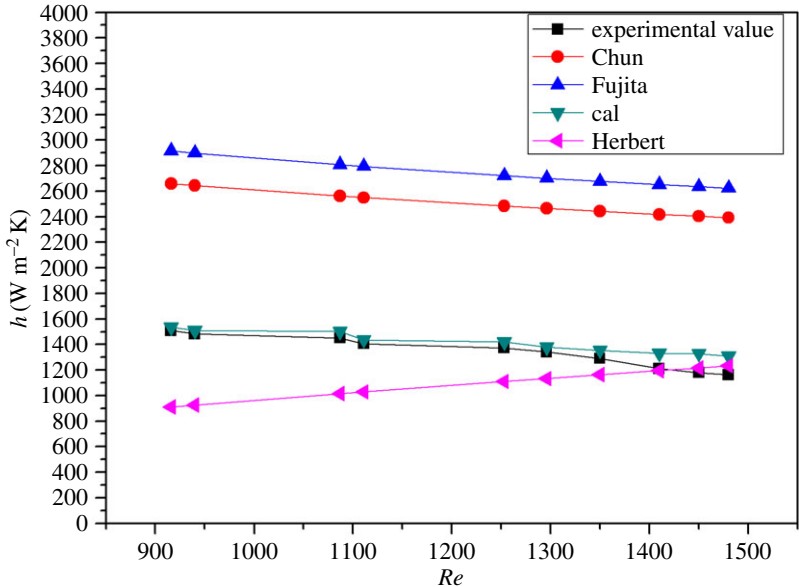

**Figure 11.** Comparison of the condensate volume with the pressure in the tube and the feed flow as variables. (*a*) $P = 50$ kPa; (*b*) $P = 60$ kPa; (*c*) $P = 70$ kPa.

**Figure 12.** Comparison between the experimental value and empirical value of falling film evaporation (laminar flow).

However, only the influence of the convective heat transfer coefficient on the overall heat transfer coefficient of the phase change on both sides of the evaporation tube is considered. Therefore, the amount of heat transfer obtained from the experimental value will be slightly smaller than the

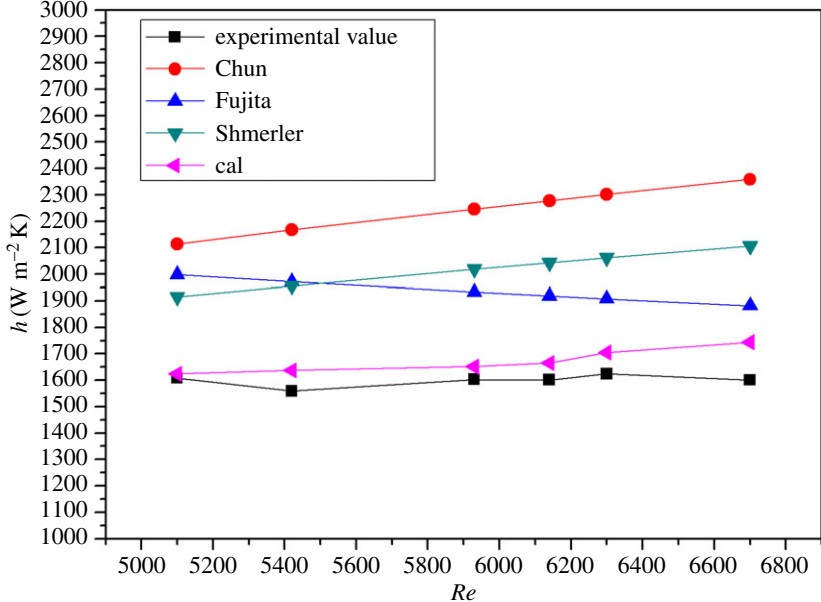

**Figure 13.** Comparison between the experimental value and empirical value of falling film evaporation (turbulent flow).

theoretically calculated value. It can be seen from the figures that the theoretical calculation value and the experimental value change trend are basically the same, and the error is below 18%.

The values of the condensate obtained by the theoretical calculation and experiment under various influencing factors were compared, and the results are shown in figures 10 and 11, respectively. From the two figures, the change experiment under the different variables is basically the same. But the theoretical calculation values are slightly larger than the experimental values, the numerical deviation of the condensate in two forms due to the thermal resistance formed by the condensate during film formation and falling. Comparing the values obtained under each operating variable, the error is within 15%. It can be seen from figures 12 and 13 that the results of this work are compared with those of Chun & Seban [13], Herbert & Sterns [14], Fujita & Ueda [15] and Shmerler & Mudawaar [24] in the laminar and transition regions. Due to the different experimental systems and experimental conditions, the error is larger. Therefore, the theoretical model for ethanol−water is closer to the experimental results than the previous model only for water, so the heat transfer model proposed in this paper is the effective and feasible model.

# 5. Conclusion

In this paper, the heat transfer coefficient correlation of the multiphase transformation inside and outside the tube is fitted by numerical simulation and verified by experiments. In order to facilitate the engineering application, the phase change between the inside and the outside of the evaporator tube is brought into the heat transfer calculation equation, and the calculation formula of the overall heat transfer coefficient of the falling film evaporator is obtained.

Calculating the overall heat transfer coefficient and heat transfer quantity in the evaporation tube, it is found that the feed flow rate and ethanol content in the evaporation tube increase, and the overall heat transfer coefficient of the equipment increases obviously. The overall heat transfer coefficient decreases when the heat transfer temperature difference between the inside and outside of the tube increases. The overall heat transfer of the equipment gradually increases when the feed flow rate, ethanol content and the temperature difference between the inside and outside of the tube increase. The change of saturated steam has no significant effect on the overall heat transfer coefficient and heat transfer quantity.

The overall heat transfer quantity and the overall heat transfer coefficient are obtained by the theoretical calculations and experiments. The error between the theoretical calculation and experiment results is within 18%, so the model proposed in this paper is effective and feasible.

Data accessibility. Data available from the Dryad Digital Repository at: https://doi.org/10.5061/dryad.v4g58s8 [25].

Authors' contributions. K.L. and M.D. carried out the laboratory work, participated in data analysis, carried out sequence alignments, participated in the design of the study and drafted the manuscript; J.F. and M.D. carried out the statistical analyses; K.L. and M.D collected field data; J.F. and M.D. conceived of the study, designed the study and coordinated the study. K.L. drafted the manuscript. All authors gave final approval for publication.

Competing interests. We declare we have no competing interests.

Funding. We received no funding for this study.

Acknowledgements. We are grateful for the support of the Hebei University of Technology.

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
