## [Reviewer comments · Royal Society Open Science]

Review History

RSOS-190135.R0 (Original submission)

Review form: Reviewer 1

Is the manuscript scientifically sound in its present form?

Yes

Are the interpretations and conclusions justified by the results?

Yes

Is the language acceptable?

Yes

Is it clear how to access all supporting data?

Yes

Do you have any ethical concerns with this paper?

No

Have you any concerns about statistical analyses in this paper?

No

Recommendation?

Accept with minor revision (please list in comments)

Comments to the Author(s)

The manuscript deals with the complex scenario of heat transfer of falling film evaporator with phase change on both the sides. The authors have developed heat transfer correlation for the evaporation within inner tube and condensation for the annular outside tube and fitted it via numerical simulation. Subsequently experiments were conducted using saturated ethanol-water mixture in the tube-side, to be heated through saturated water vapor in the outer annular section. Results obtained are encouraging as evident from Figs 8 to 11 of the manuscript.

Literature search is reasonably exhaustive where classical references as well as recent important publications are duly quoted. However following points need to be addressed for betterment of the manuscript:

1. Its better to use the standard symbols like "t" for time and "T" for temperature to reduce chance of confusion.
2. Similarly, most of the scientific literature uses "k" as the thermal conductivity and the symbol used in manuscript " λ_{w} " generally refers latent heat in standard literature.
3. It is reported that the maximum error between theoretical prediction and experimentally observed findings are within 18%. It will be appropriate to have some comparative account of the experimental observation with other theoretical models available in the open literature in order to judge performance of the developed model.
4. Authors have considered modeling of the liquid film using the 2D Cartesian coordinate system, however the geometry of falling film evaporator clearly indicates it would be appropriate to consider 3-dimensional polar coordinates.

Review form: Reviewer 2

Is the manuscript scientifically sound in its present form?

Yes

Are the interpretations and conclusions justified by the results?

Yes

Is the language acceptable?

Yes

Is it clear how to access all supporting data?

Not Applicable

Do you have any ethical concerns with this paper?

No

Have you any concerns about statistical analyses in this paper?

No

Recommendation?

Accept with minor revision (please list in comments)

Comments to the Author(s)

In this manuscript, the authors studied the heat transfer of falling film evaporation. Temperature distributions were calculated by empirical heat transfer equations. The heat transfer coefficients were estimated accordingly, and the relationship between the coefficients and Reynolds number and Prandtl number was built via correlation.

Finally, experiments were conducted where mixture of ethanol and water at different ratios flowed downward inside of the evaporator under various pressure. The measured heat transfer amounts was comparable to the values calculated from the model, which further supports the author's theoretical analysis. This work provides valuable information for a common scenario (water-ethanol separation), and the developed correlations would serve as a good reference for people in the field. I recommend acceptance of this work for publication once the following issues being addressed.

1. Some English grammars should be corrected. For example, Page 1, line 31, "have great significance" should be changed to "has great significance"
2. On Page 3, equation 7, please define lamda
3. When listing the equations of temperature distribution, the authors used "t" to represent temperature. Normally "T" is commonly used character to represent temperature.
4. On Page 3, last sentence, "Equation 8 can be...." should be changed to "Equation 9 can be....".
5. The numerical values of boundary conditions should be provided.
6. In Figure 8 - 11, I suggest using an identical range on all Y axis in order to visualize the change of heat transfer quantity under different conditions. Also, the authors should dig more info out of their data. For example, the heat transfer quantity varies with experimental conditions, what is the mechanism behind this observation?

Decision letter (RSOS-190135.R0)

01-Apr-2019

Dear Dr Li

On behalf of the Editors, I am pleased to inform you that your Manuscript RSOS-190135 entitled "Establishment of the Falling Film Evaporation Model and Correlation of the Overall Heat Transfer Coefficient" has been accepted for publication in Royal Society Open Science subject to minor revision in accordance with the referee suggestions. Please find the referees' comments at the end of this email.

The reviewers and handling editors have recommended publication, but also suggest some minor revisions to your manuscript. Therefore, I invite you to respond to the comments and revise your manuscript.

- Ethics statement

If your study uses humans or animals please include details of the ethical approval received, including the name of the committee that granted approval. For human studies please also detail

whether informed consent was obtained. For field studies on animals please include details of all permissions, licences and/or approvals granted to carry out the fieldwork.

- Data accessibility

If you wish to submit your supporting data or code to Dryad (<http://datadryad.org/>), or modify your current submission to dryad, please use the following link:
<http://datadryad.org/submit?journalID=RSOS&manu=RSOS-190135>

- Competing interests

- Authors' contributions

- Acknowledgements

- Funding statement

Because the schedule for publication is very tight, it is a condition of publication that you submit the revised version of your manuscript before 10-Apr-2019. Please note that the revision deadline

will expire at 00.00am on this date. If you do not think you will be able to meet this date please let me know immediately.

If your manuscript is newly submitted and subsequently accepted for publication, you will be asked to pay the article processing charge, unless you request a waiver and this is approved by

Royal Society Publishing. You can find out more about the charges at <http://rsos.royalsocietypublishing.org/page/charges>. Should you have any queries, please contact openscience@royalsociety.org.

on behalf of Professor R. Kerry Rowe (Subject Editor)
openscience@royalsociety.org

Associate Editor Comments to Author:

Please carefully address both the reviewers' comments and provide a point by point response. Reviewer 2 has suggested that the quality of English needs to be improved. A number of language polishing services are available for authors whose first language is not English. <https://royalsociety.org/journals/authors/language-polishing/>

Authors whose papers are returned on language grounds must provide evidence that a professional language editing service or a native speaker of English have assisted in preparing a revised manuscript. Evidence such as a certificate of editing or a signed letter from a native speaker of English would be acceptable.

Reviewer comments to Author:

Reviewer: 1

Comments to the Author(s)

The manuscript deals with the complex scenario of heat transfer of falling film evaporator with phase change on both the sides. The authors have developed heat transfer correlation for the evaporation within inner tube and condensation for the annular outside tube and fitted it via numerical simulation. Subsequently experiments were conducted using saturated ethanol-water mixture in the tube-side, to be heated through saturated water vapor in the outer annular section. Results obtained are encouraging as evident from Figs 8 to 11 of the manuscript. Literature search is reasonably exhaustive where classical references as well as recent important publications are duly quoted. However following points need to be addressed for betterment of the manuscript:

1. Its better to use the standard symbols like "t" for time and "T" for temperature to reduce chance of confusion.
2. Similarly, most of the scientific literature uses "k" as the thermal conductivity and the symbol used in manuscript " λ_{w} " generally refers latent heat in standard literature.
3. It is reported that the maximum error between theoretical prediction and experimentally observed findings are within 18%. It will be appropriate to have some comparative account of the experimental observation with other theoretical models available in the open literature in order to judge performance of the developed model.
4. Authors have considered modeling of the liquid film using the 2D Cartesian coordinate system, however the geometry of falling film evaporator clearly indicates it would be appropriate to consider 3-dimensional polar coordinates.

Reviewer: 2

Comments to the Author(s)

In this manuscript, the authors studied the heat transfer of falling film evaporation. Temperature distributions were calculated by empirical heat transfer equations. The heat transfer coefficients were estimated accordingly, and the relationship between the coefficients and Reynolds number and Prandtl number was built via correlation.

Finally, experiments were conducted where mixture of ethanol and water at different ratios flowed downward inside of the evaporator under various pressure. The measured heat transfer amounts was comparable to the values calculated from the model, which further supports the author's theoretical analysis. This work provides valuable information for a common scenario (water-ethanol separation), and the developed correlations would serve as a good reference for people in the field. I recommend acceptance of this work for publication once the following issues being addressed.

1. Some English grammars should be corrected. For example, Page 1, line 31, "have great significance" should be changed to "has great significance"
2. On Page 3, equation 7, please define lamda
3. When listing the equations of temperature distribution, the authors used "t" to represent temperature. Normally "T" is commonly used character to represent temperature.
4. On Page 3, last sentence, "Equation 8 can be...." should be changed to "Equation 9 can be....".
5. The numerical values of boundary conditions should be provided.
6. In Figure 8 - 11, I suggest using an identical range on all Y axis in order to visualize the change of heat transfer quantity under different conditions. Also, the authors should dig more info out of their data. For example, the heat transfer quantity varies with experimental conditions, what is the mechanism behind this observation?

Author's Response to Decision Letter for (RSOS-190135.R0)

See Appendix A.

Decision letter (RSOS-190135.R1)

11-Apr-2019

Dear Dr Li,

I am pleased to inform you that your manuscript entitled "Establishment of the Falling Film Evaporation Model and Correlation of the Overall Heat Transfer Coefficient" is now accepted for publication in Royal Society Open Science.

on behalf of Professor R. Kerry Rowe (Subject Editor)
openscience@royalsociety.org

Follow Royal Society Publishing on Twitter: [@RSocPublishing](https://twitter.com/RSocPublishing)
Follow Royal Society Publishing on Facebook:
<https://www.facebook.com/RoyalSocietyPublishing.FanPage/>
Read Royal Society Publishing's blog: <https://blogs.royalsociety.org/publishing/>

Appendix A

List of Responses

Dear Editors and Reviewers:

Thank you very much for your letter and for the reviewers' comments concerning our manuscript entitled "Establishment of the Falling Film Evaporation Model and Correlation of the Overall Heat Transfer Coefficient" (ID: RSOS-190135). Those comments are all valuable and very helpful for revising and improving our paper, as well as the important guiding significance to our researches. We have studied comments carefully and have made correction which we hope meet with approval. Revised portion is marked in red in the paper. The main corrections in the paper and the responds to the reviewer's comments are as flowing:

Responds to the reviewer's comments:

Reviewer 1:

1. Response to comment: Its better to use the standard symbols like "t" for time and "T" for temperature to reduce chance of confusion.

Response: We changed all the temperature symbols to "T".

2. Response to comment: Similarly, most of the scientific literature uses "k" as the thermal conductivity and the symbol used in manuscript " λ_w " generally refers latent heat in standard literature.

Response: We changed all the thermal conductivity symbols to "k".

3. Response to comment:It is reported that the maximum error between theoretical prediction and experimentally observed findings are within 18%. It will be appropriate to have some comparative account of the experimental observation with other theoretical models available in the open literature in order to judge performance

of the developed model.

Response: We have made correction according to the Reviewer's comments. We compared the theoretical models obtained by previous researchers.

4. Response to comment: Authors have considered modeling of the liquid film using the 2D Cartesian coordinate system, however the geometry of falling film evaporator clearly indicates it would be appropriate to consider 3-dimensional polar coordinates.

Response: First of all, thanks to the reviewer's suggestion, we studied the falling film heat transfer coefficient in a vertical smooth tube, 3D vertical tube is axisymmetric, the circumferential parameter has little effect on the falling film process and can be ignored. So we simplifying the falling film process appropriately, and the simplified model calculation value is not much different from the experimental results. In the process of research, we will analyze the differences between 3D modeling and 2D modeling and their respective advantages in the subsequent study.

Special thanks to you for your good comments.

Reviewer 2:

1. Response to comment: Some English grammars should be corrected. For example, Page 1, line 31, "have great significance" should be changed to "has great significance"

Response: We changed "have great significance" to "has great significance". And we found someone whose native language is English to help with the revision of the manuscript and to amend the question you asked.

2. Response to comment: On Page 3, equation 7, please define lamda

Response: According to the opinions of the first reviewer, We changed all the thermal conductivity symbols to "k". On Page 3, equation 7, k_0 is the thermal conductivity of condensate.

3. Response to comment: When listing the equations of temperature distribution, the authors used "t" to represent temperature. Normally "T" is commonly used character to represent temperature.

Response: We changed all the temperature symbols to "T".

4. Response to comment: On Page 3, last sentence, "Equation 8 can be...." should be changed to "Equation 9 can be....".

Response: We are very sorry for our incorrect writing. We changed "Equation 8 can be...." to "Equation 9 can be....".

5. Response to comment: The numerical values of boundary conditions should be provided.

Response: We have made correction according to the Reviewer's comments. We have provided all the numerical values of boundary conditions.

6. Response to comment: In Figure 8 - 11, I suggest using an identical range on all Y axis in order to visualize the change of heat transfer quantity under different conditions. Also, the authors should dig more info out of their data. For example, the heat transfer quantity varies with experimental conditions, what is the mechanism behind this observation?

Response: We have made correction according to the Reviewer's comments. We dig more info out of our data.

Special thanks to you for your good comments.

We tried our best to improve the manuscript and made some changes in the manuscript. These changes will not influence the content and framework of the paper. And here we did not list the changes but marked in red in the revised paper.

We appreciate for Editors/Reviewers' warm work earnestly and hope that the correction will meet with approval.

Once again, thank you very much for your comments and suggestions.